# Correlating Stability-Indicating Biochemical and Biophysical Characteristics with In Vitro Cell Potency in mRNA LNP Vaccine

**DOI:** 10.3390/vaccines12020169

**Published:** 2024-02-07

**Authors:** Xin Tong, Jessica Raffaele, Katrina Feller, Geethanjali Dornadula, James Devlin, David Boyd, John W. Loughney, Jon Shanter, Richard R. Rustandi

**Affiliations:** 1Analytical Research & Development, Merck & Co., Inc., Rahway, NJ 07065, USAkatrina.fritts@merck.com (K.F.); geethanjali_dornadula@merck.com (G.D.); john_loughney@merck.com (J.W.L.); richard_rustandi@merck.com (R.R.R.); 2Process Research & Development, Merck & Co., Inc., Rahway, NJ 07065, USA; david_boyd@merck.com (D.B.); jon_shanter@merck.com (J.S.)

**Keywords:** mRNA vaccines, lipid nanoparticles, potency, stability

## Abstract

The development of mRNA vaccines has increased rapidly since the COVID-19 pandemic. As one of the critical attributes, understanding mRNA lipid nanoparticle (LNP) stability is critical in the vaccine product development. However, the correlation between LNPs’ physiochemical characteristics and their potency still remains unclear. The lack of regulatory guidance on the specifications for mRNA LNPs is also partially due to this underexplored relationship. In this study, we performed a three-month stability study of heat-stressed mRNA LNP samples. The mRNA LNP samples were analyzed for their mRNA degradation, LNP particle sizes, and mRNA encapsulation efficiency. In vitro cell potency was also evaluated and correlated with these above-mentioned physiochemical characterizations. The mRNA degradation–cell potency correlation data showed two distinct regions, indicating a critical cut-off size limit for mRNA degradation. The same temperature dependence was also observed in the LNP size–cell potency correlation.

## 1. Introduction

Recent success stories about COVID-19 vaccine development from BioNtech/Pfizer and Moderna using mRNA technology have triggered many other vaccine candidates in clinical trials using similar technology [1,2]. One major component of an mRNA vaccine is the lipid nanoparticle (LNP). LNPs consist of lipids that encapsulate mRNA strands that encode a specific target protein and serve as mRNA protectants, delivery systems into cells, and adjuvants [3,4,5]. The current mRNA vaccines on the market usually have similar compositions of LNPs. The LNP normally contains a cationic lipid, a polyethylene glycol (PEG)–lipid conjugate, cholesterol, and a zwitterionic helper phospholipid. The cationic lipid interacts with negatively charged mRNA during LNP encapsulation. The inclusion of PEGylated lipids in LNP formulations results in a “steric stabilization” outcome wherein a hydrophilic layer is created on the particle surface. This layer can prevent particle aggregation and reduce uptake by macrophages of the mononuclear phagocyte system (MPS) [6]. 1,2-distearoyl-sn-glycero-3-phosphocholine (DSPC), also used in the COVID-19 vaccines, can stabilize the LNP structure with its high melting temperature [7]. Cholesterol can enhance particle stability by modulating membrane integrity and rigidity [8,9,10].

One of the mRNA-LNP vaccines in clinical development is against the respiratory syncytial virus (RSV) [11,12]. RSV belongs to the *pneumoviridae* family, which causes lower and upper respiratory disease, resulting in mortality and hospitalization in infant and adult populations. Licensed RSV-preventive antibody products (Beyfortus and Synagis) are available for infants and young children. As of 2023, two RSV vaccines (Arexvy and Abrysvo) are approved for people aged 60 years and older. Due to the seasonality and ongoing severity of RSV infections, additional RSV vaccine candidates are needed. The majority of RSV developmental vaccines are targeted against the fusion (F) protein, conserved among clinical isolates. RSV F-protein belongs to the type 1 fusion glycoprotein, and it has been established that it could have two conformations between metastable pre-fusion and stable post-fusion. Most neutralizing antibodies elicited by natural infection in humans are predominantly directed toward the pre-fusion form of RSV F [13,14].

Various particle characteristics, such as size, shape, surface charge (zeta potential), solubility, surface modifications, and mode of administration, can influence the distribution of nanoparticles in the human body [15]. Therefore, during the development of this mRNA-LNP RSV vaccine, several quality attributes were assessed to understand the product profile stability. It is expected that mRNA size affects protein production as well as the loss of capping and the polyA tail in mRNA [16,17]. Multiple publications have demonstrated that LNP particle size affects cell uptake [6,18,19]. The degree of mRNA encapsulation into LNP, called encapsulation efficiency, could also dramatically affect protein translation [3,4]. Finally, lipid degradation may affect the LNP stability as well [20].

## 2. Materials and Methods

The mRNA LNP materials were developed and manufactured in-house, similar to our previous work [21]. The mRNA LNPs contain a cationic lipid MC3 (also known as DLin-MC3-DMA), DSPC, cholesterol, and poly (ethylene glycol) 2000-dimyristoylglycerol (PEG2000-DMG). The mRNA LNP samples were placed in −70 °C, −20 °C, 4 °C, 25 °C, 37 °C, and 45 °C chambers for 3 months. Aliquots were tested for physiochemical properties and in vitro potency at the corresponsive time points (Appendix A). All the other materials used are detailed in the corresponding methods.

### 2.1. mRNA Size-Based Assay

Raffaele et al. [22] have recently published a study elucidating the rate of mRNA degradation at various temperatures using a microchip capillary electrophoresis method. Similarly, the mRNA-LNP samples were first diluted to a mRNA concentration of 100 μg/mL in a solution of 10% *w*/*v* Brij^®^ 58 in formamide. Then, samples were further diluted in formamide and 5 μL of 10X sample buffer from the RNA reagent kit (Perkin Elmer, Waltham, MA, USA) for a final total sample volume of 50 μL (10 μg/mL mRNA final concentration). The final formamide concentration in the sample was always >80%. All final sample solutions were heated in a 70 °C heating block for 10 min, then cooled on ice for at least 5 min. Samples were transferred to a 96-well plate before testing on the LabChip GXII Touch (Perkin Elmer, Waltham, MA, USA). The RNA lab chip was prepared as described in the RNA Assay Quick Guide provided by Perkin Elmer without any modifications. Each component of the main peak and fragments was calculated as a percentage of the total peak area. The percent main peak area was reported as the % mRNA integrity.

### 2.2. LNP Particle Size Assay

The Malvern Zetasizer Nano ZS (Malvern Panalytical, Malvern, UK) is used to estimate the average particle size of the LNP in terms of hydrodynamic diameter (Dh) through the technique of batch-mode Dynamic Light Scattering (DLS). DLS uses a laser to illuminate particles in a solution and then examines the changes in the intensity of the scattered light over time as a result of the Brownian motion of illuminated particles. The correlation of the scattered light intensity over time to the intensity at time zero results in an exponential decay curve, or correlation function. The rate of decay correlation function, concerning time, is much faster for smaller particles than larger particles. Therefore, this correlation function, along with the Stokes–Einstein equation, can be used to calculate the mean hydrodynamic size of the sample particles. The Zetasizer software (Version 7.13) uses algorithms to extract decay rates for several size classes to facilitate the de-convolution of size distribution.

### 2.3. Encapsulation Efficiency Assay

The encapsulation efficiency (%EE) of LNPs is calculated by Equation (1). In detail, “undisrupted LNP mRNA concentration (conc.)” was determined by testing the free mRNA in the untreated LNP samples. To test the overall free and encapsulated mRNA concentration, the LNP samples were treated with 0.1% Triton X-100 buffer (Thermo Fisher, Waltham, MA, USA) and tested as the “disrupted LNP mRNA conc”. Samples were detected by the commercial RiboGreen RNA kits (Molecular Probes, Eugene, OR, USA) following their procedures. The RiboGreen reagent is a fluorescent nonspecific nucleic acid-binding dye that is excited at a wavelength of 485–488 nm and emits fluorescence at 525–530 nm. mRNA concentration was determined by comparing fluorescence values against a ribosomal RNA (rRNA) reference standard provided in the kit.
(1)$$\%EE=\frac{\left[Disrupted\;LNP\;mRNA\;Conc.\right]-[Undisrupted\;LNP\;mRNA\;Conc.]}{\left[Disrupted\;LNP\;mRNA\;Conc.\right]}$$

### 2.4. In Vitro Cell-Based Potency Assay

Li et al. [23] recently published the development and qualification of the cell-based potency assay for the mRNA-LNP RSV vaccine. Briefly, HepG2 cells are seeded at a density of 1.2 × 10^5^ cells/well in 96-well plates in EMEM Media + 2% Heat-inactivated FBS and 1% Penicillin Streptomycin and incubated at 37 °C, 5% CO_2_ for 24 ± 4 h. The 2-fold-diluted reference standard and mRNA LNP samples in Opti-MEM medium at the highest 800 ng/mL concentrations were transfected and incubated in the plates at 37 °C, 5% CO_2_ overnight. After 16–18 h of post-transfection incubation, the medium was removed from the cell plates. The cells were fixed with 3.7% Formaldehyde for 20 min at RT, permeabilized with PBS + 1% Triton X-100, and probed with the anti-RSV antibody RBMD21Nov2014_3D10 (internally produced) at one µg/mL for one hour with moderate shaking on a rotator at room temperature. The cell plates were washed with PBS + 0.05% Tween 20 and incubated with IRDye 680RD goat α-human (Li-COR) diluted 100-fold from 1 mg/mL stock to 10 µg/mL for 2 h at room temperature. Plates were washed with 1% PBS three times, and plates were Imaged using a SpectraMax i3X MiniMax instrument (Molecular Devices, San Jose, CA, USA) to acquire both fluorescent and transmitted light images. The acquired images were subsequently analyzed by the MiniMax image analysis algorithm. The relative potency values for each sample were determined based on the reference standard using SoftMax Pro GxP software (Version 6.5.1). Relative potencies of mRNA LNP samples were calculated using 4-parameter logistic fits based on reference standard and dilution factors.

## 3. Results

In this study, we describe the correlation between some critical biochemical and biophysical attributes with in vitro cell-based potency assays for mRNA-LNP RSV vaccines. The correlation was studied by monitoring heat-stressed mRNA-LNP samples. These samples were placed in −70 °C, −20 °C, 4 °C, 25 °C, 37 °C, and 45 °C chambers for 3 months. The stability results for individual characterizations are provided below or in the Appendix A. 

### 3.1. The Stability of mRNA Correlates with In Vitro Potency

The stability of mRNA was monitored using microchip size-based capillary electrophoresis to monitor its size (Figure 1A). Correlating with the in vitro cell-based relative potency (Figure 1B), the correlation results are shown in Figure 1C.

mRNA integrity remained stable for three months at −20 °C and 4 °C, with % intact mRNA values above 60%. As for the other elevated conditions, the mRNA integrity showed a temperature-related trend. The % intact mRNA decreased more rapidly as the storage temperature increased. A similar trend can also be found in the cell-based relative potency results. As the sample storage temperature increased from 25 °C to 45 °C, the % potency dropped more quickly. An interesting phenomenon was observed for LNP samples stored at 4 °C, where the % potency decreased more slowly than at −20 °C. It is worth mentioning that while there is a general similarity in the overall potency trend between 4 °C and −20 °C, the samples demonstrate dissimilar correlations, as shown in Figure 1C.

The data indicate two distinct regions from the correlation. As the % intact mRNA reduced to 60%, the cell-based potency also decreased, demonstrating a gradual positive linear correlation (yellow circle with the yellow line). Surprisingly, when the % intact mRNA reduced below 45%, there was a sharp drop in the cell potency to almost 0% (red circle). Another interesting observation is that all samples at −20 °C and −70 °C showed no decrease in intact mRNA, but the potency continued to decrease, resulting from storage time (purple line). This observation suggests that other than mRNA integrity, other underlying factors may influence the samples’ potency.

### 3.2. Particle Size Partially Correlates with In Vitro Potency in Certain Conditions

The same set of samples was analyzed by DLS to measure the LNP particle size (Appendix A) and correlated with cell potency, as shown in Figure 2A. The results indicate that most of the LNP particle sizes fell in the 95–125 nm range, except for three samples stored at −20 °C (shown in purple and pointed out by the red arrows). They had particle sizes of 132 nm, 141 nm, and 147 nm. We further extrapolated all the frozen samples’ particle size data (−20 °C and −70 °C) and plotted them against potency, as shown in Figure 2B. It is clear that there is a dependence between LNP size and potency. As the LNP size increases, potency decreases, with the potency decreasing to approximately 50% when the LNP particle size is at ~130 nm.

### 3.3. Encapsulation Efficiency Does Not Correlate with In Vitro Potency

Another critical attribute tested for the LNP samples is the encapsulation efficiency (Appendix A). No significant differences were observed in the encapsulation efficiency tested for LNP samples. Even at elevated conditions, the majority of %EE tested remained stable over time. We did not see any significant changes for samples stored at 4 °C and 25 °C based on the findings from this study. Hence, we do not anticipate any change in the −20 °C either. Unlike the other two physical characterizations, there was no clear correlation between the %EE and the cell potency, as shown in Figure 3.

## 4. Discussion

The relationship between the physical attributes and the potency of mRNA LNPs remains a huge interest in this field. A clear relationship can help the industry justify the specifications set for these assays and provide confidence for the regulatory agencies to provide more guidance. Moreover, potency, as a key indicator for the stability of vaccines, creates analytical challenges and burdens. The development process can be expedited with a more elucidated correlation between the potency and physiochemical characterizations of LNPs, which are easier to measure.

The LNP vaccines are intended to be used as frozen liquid and stored at −70 °C. For this short-term stability study, the conditions (−20 °C, 4 °C, 25 °C, 37 °C, and 45 °C) are considered elevated conditions. In the elevated conditions, the LNPs were supposed to have accelerated degradation. It is commonly acknowledged in the industry that mRNA integrity is a critical attribute to monitoring mRNA vaccines’ functionality and stability [24,25]. Our data show that potency decreases with decreasing intact mRNA linearly until there is a sudden drop in potency at ~60% intact. The potency appears to decrease to 50% at 65% intact mRNA. Although more studies are needed in the future, we suspect it reaches a critical point in protein production at 65% mRNA integrity. LNPs stored at −20 °C and −70 °C show no loss in intact mRNA as expected since mRNA fragmentation involves a bond-breaking process; hence, in a frozen state, there is no/little bond breaking due to insufficient energy to reach the transition state. Despite stable, intact mRNA, the potency decreases for samples stored at −20 °C, suggesting another mechanism is involved, resulting in a decrease in potency other than intact mRNA.

One such proposed mechanism is based on particle size. The majority of the samples analyzed were in the size range of 95–125 nm, where any observed lower potency (<50%) is most likely due to the degraded intact mRNA size from those samples. However, as the LNP size increases beyond ~130 nm, the potency decreases to less than 50% (Figure 2B). Internal research showed a similar correlation between LNP size and potency, where the researchers fractionated the LNP size by HPSEC, and each fraction was measured by DLS and potency. Their results show that the LNP size range of 80 to 130 nm gives good potency, while the potency drops significantly with LNP outside this range (size <80 nm or >130 nm). We suspect the observed phenomenon is due to the limitation of the in vitro model, where phagocytes and some pinocyte entrances are unavailable [26]. Phagocytes, involving macrophages and monocytes, can take large particles. It is hard for larger-sized (>130 nm) LNPs to enter cells in vitro, causing a drop in potency.

However, this size-dependent immune response might not occur in in vivo models. Recently, Hassett et al. published an in vivo mouse and NHP study evaluating the impact of mRNA-LNP size on the immunogenicity of a cytomegalovirus (CMV) vaccine [19]. The results suggest that the immune response is affected for smaller LNP sizes (<70 nm), while there is no statistical difference for larger sizes from 81 to 146 nm. It is likely that phagocytosis and pinocytosis exist in vivo. Hence, there is no size dependence for generating immunogenicity. This suggests that establishing in vitro–in vivo correlation is highly valuable for mRNA vaccine evaluation.

## 5. Conclusions

Our research studied the correlation between mRNA LNPs’ physiochemical properties and their in vitro potency under various storage conditions. The results suggest that mRNA integrity and particle size, among other factors, can influence in vitro potency. Although in vivo studies are needed for more comprehensive evaluations, establishing such a relationship can help expedite the drug development process for similar mRNA vaccines. The findings from this study can aid in justifying assay specifications and providing regulatory agencies with more guidance.

## Figures and Tables

**Figure 1 vaccines-12-00169-f001:**
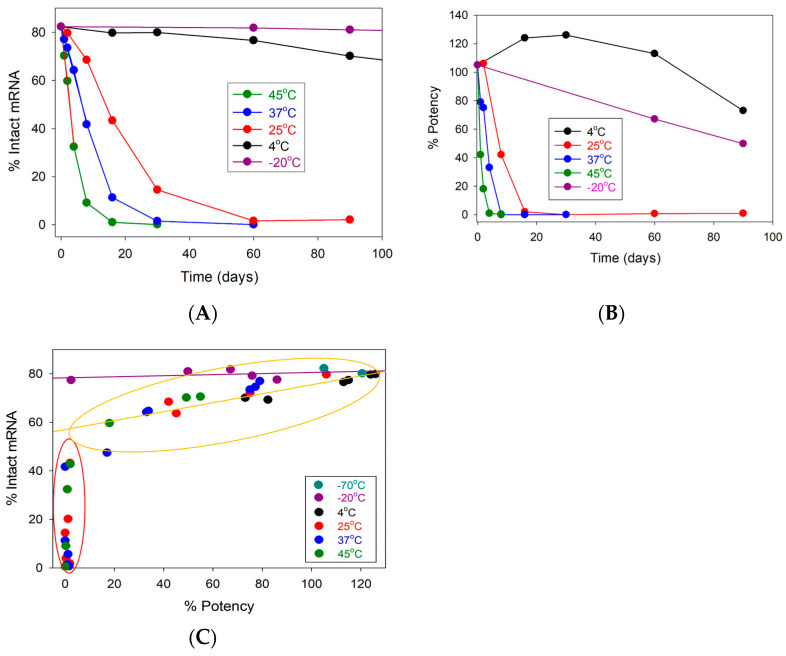
(**A**) mRNA integrity (%intact mRNA, *y*-axis) was plotted against time (days, *x*-axis). (**B**) Cell-based relative potency (*y*-axis) was plotted against time (days, *x*-axis). (**C**) Correlations between %intact mRNA (integrity) in the *y*-axis and the cell-based relative potency in the *x*-axis. Two colored circles (yellow and red) indicate two distinguished subsets. Two colored solid lines (yellow and purple) indicate positive and zero correlations, respectively. Each data point represents the average result tested at a single time point within their respective storage condition. For in vitro potency and mRNA integrity testing, samples were tested in duplicate (*n* = 2).

**Figure 2 vaccines-12-00169-f002:**
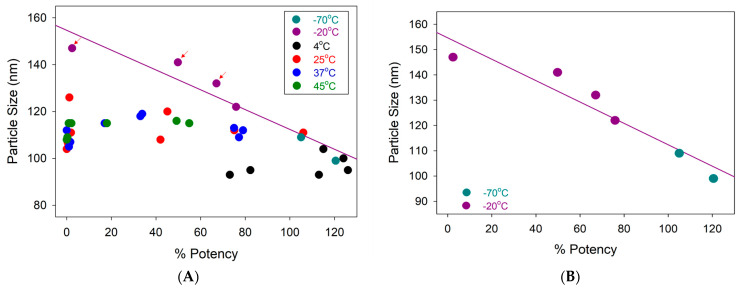
(**A**) Correlations between the LNP particle size in the *y*-axis and the cell-based relative potency in the *x*-axis with all the data points. (**B**) Correlations between the LNP particle size and the cell-based relative potency with only frozen samples (−20 °C and −70 °C). Each data point represents the average result tested at a single time point within its respective storage condition. For in vitro potency and particle size testing, samples were tested in duplicate (*n* = 2).

**Figure 3 vaccines-12-00169-f003:**
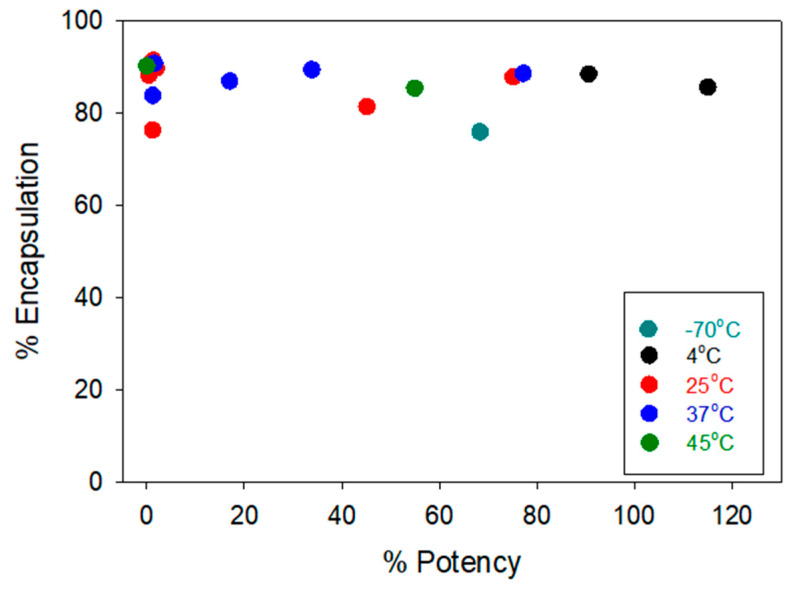
Correlations between the LNP encapsulation efficiency (%EE) in the *y*-axis and the cell-based relative potency in the *x*-axis. Each data point represents the average result tested at a single time point within its respective storage condition. For in vitro potency testing, samples were tested in duplicate (*n* = 2). For encapsulation efficiency testing, samples were tested in quadruplicate (*n* = 4).

## Data Availability

Data are contained within the article and Appendix A.

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
