# Peer review of "Correlating Stability-Indicating Biochemical and Biophysical Characteristics with In Vitro Cell Potency in mRNA LNP Vaccine"

_vaccines, 2024, doi:10.3390/vaccines12020169_

Round 1

Reviewer 1 Report

Comments and Suggestions for Authors

The premise behind the work described in this manuscript is interesting, and as the authors state, valuable to inform industry and regulatory agencies guidance on the production of vaccines.

Line 42. Give more detail. If there is a vaccine available, why are more in development?

The issue with this manuscript is its brevity. While concise writing is commendable it should not be at the expence of explanation. As an example, in the abstract it was stated that a three-month stability assay was carried out, but there was no mention of storage conditions in the methods. Similarly, in the results, the graphs were not explained well. In Figure 1 for example, where the samples in duplicate, triplicate? There were more points on the graph than in the key. In general I found Figure 1 very unclear. The Legend needed much more information. The statement at line 125 is dissucult to undestand. Why were thyere differences in potency between (presumed) triplicates from the same storage conditions.

In summary, Figure 1 needs a total re-work.

Further questions/comments.

How was the encapsulation efficiency calculated?

Line 130 has a comment about storage for longer than one month. Again, define more precisely the experimental protocol.

Figure 2A, what are all the data points? How many per condition?

Figure 3, again define the number of replicates in the legend.

Discussion. Expand somewwhat. In particular:

Line 159-160. Speculate on why this is.

The discussion needed an overall conclusion. Why are these findings important? What needs to be done in the future?

Comments on the Quality of English Language

English is generally good. Minor points.

Line 10. heated surge is an odd phrase. Not really scientific.

Line 36. Define DSPC.

Line 46. Hyphenate pre-fusion as per post-fusion.

Line 50. Use "such as targeting". Like means similar to.

Line 180. You use italics here but not elsewhere. Be consistent.

Author Response

We thank the reviewer for their feedback. Your comments helped us improve the quality of this work. Please see the attachment for detailed responses. 

Reviewer 2 Report

Comments and Suggestions for Authors

Tong et al., have submitted the study entitled “Correlating Stability-Indicating Biochemical and Biophysical Characteristics with in-vitro Cell Potency in mRNA LNP Vaccine.”

The authors have studied the mRNA-LNP physicochemical properties and correlated their effects on cell-based potency. The authors used the different temperatures to assess the different parameters for 3 months.

Although it is a short communication, it should reach the expectations.

This reviewer has the following questions and suggestions.

Major

Details about mRNA-LNP are missing? Which mRNA is used or LNP composition?

Why is the 3 months fixed? Discussion is lacking on this aspect.

There is no information about the number of experiments conducted and the type of statistics applied. The results look very vague.

What is the correlation method used?

Figure 3: Why %EE is missing for -20 degrees, which is a very important aspect for mRNA-LNPs.

Minor

Line 40: “pneumoviridae” should be in italics.

"licensed RSV vaccine" What is that? please mention this in the parenthesis.

Line 78: " drug product" ----- LNP

Line 160: "Drug substance" ---- should be "LNP"

The last paragraph of the discussion is not in a flow or in an expected way.

Author Response

We thank the reviewer for their detailed feedback. Your comments helped us improve the quality of this work. Please see the attachment for detailed responses.

Round 2

Reviewer 1 Report

Comments and Suggestions for Authors

The authors have done a commendable job in addressing my comments, and the manuscript is now much clearer. I recommend acceptance as it now stands. 

Author Response

We thank the reviewer for their continued support of our work. The reviewer agrees to accept the manuscript as the current version. Therefore, no response is needed. Thanks. 

Reviewer 2 Report

Comments and Suggestions for Authors

The authors have reasonably revised the manuscript. 

The revised manuscript is acceptable for publication. 

Author Response

We thank the reviewer for their continued efforts. Since the reviewer agrees that the revised manuscript is acceptable for publication, no response is needed. 

Thank you for your time!